# Preparation and Performance of Thickened Liquids for Patients with Konjac Glucomannan-Mediated Dysphagia

**DOI:** 10.3390/molecules27072194

**Published:** 2022-03-28

**Authors:** Wen Zhang, Xuening Ren, Lele Zhang, Jianghu Chen

**Affiliations:** School of Food and Biological Engineering, Shaanxi University of Science & Technology, Xi’an 710021, China; rxn1013@163.com (X.R.); zhanglele1225@163.com (L.Z.); cjh526395@163.com (J.C.)

**Keywords:** Konjac glucomannan, swallowing disorders, thickening component, rheological behavior

## Abstract

The present study sought to characterize the rheological and thickening properties of Konjac glucomannan (KGM) and prepare thickening components for special medical purposes using KGM and maltodextrin as the primary raw materials and guar gum (GG), xanthan gum (XG), locust bean gum (LBG), and carrageenan (KC) as the supplemented materials. The formulation and preparation processes were optimized through single factor experiments taking sensory evaluation as an indicator. The results confirm that KGM had excellent thickening performance, reaching about 90 times its own mass. The optimal formulation process of the thickening components based on KGM was as follows: the mass concentration of the compound thickener (KGM/GG/XG/LBG/KC = 13:2:2:2:1) was 5.0–7.0 mg/mL; the maltodextrin concentration was 10.0 mg/mL; the brewing temperature of the thickening component was 60 °C with no restriction on consumption time. The rheology test results revealed that the thickening components had shear thinning characteristics, which could provide three different thickening effects of nectar-thick level (350 mPa·s), honey-thick level (1250 mPa·s), and pudding-thick level (1810 mPa·s) suitable for people with different degrees of chewing disorders. Overall, this study provides a theoretical basis and technical reference for KGM as a dietary nutrition support for patients with dysphagia.

## 1. Introduction

Swallowing disorders, defined as difficulties in eating and swallowing due to neural or muscular control abnormalities (Figure 1), often occur in the elderly population and in people with specific diseases, such as stroke and neurodegenerative and Alzheimer’s diseases, leaving them more prone to major health problems, such as malnutrition and aspiration pneumonia [1,2]. It is estimated that 8% of the worldwide population experiences difficulty in eating regular food and drinking regular fluids due to dysphagia [3]. Electrical stimulation treatment is an effective clinical approach for short-term recovery in patients with swallowing disorders [4]. However, modifying the food texture and liquid thickness for treating such disorders has attracted much attention. Generally, liquids are thickened to slow their transmission through the mouth and throat stages of swallowing and to avoid material inhalation into the airway [3], giving muscle a longer reflex response time [5] and reducing the risk of aspiration during swallowing [6]. A survey by 145 speech–language pathologists claimed that 84.8% of the respondents believed that nectar-thick is a very effective measure for patients with dysphagia [7]. Clavé P et al. [8] concluded that both nectar-thick and pudding-thick liquids reduced aspiration in patients with diagnoses of brain injury and neurodegenerative diseases.

The National Dysphagia Diet (NDD) guideline has categorized viscosity into three groups, i.e., nectar-thick (51–350 mPa·s), honey-thick (351–1750 mPa·s), and spoon-thick (>1750 mPa·s) foodstuffs. However, the NDD classification has not considered other factors affecting fluid flow [9]. The viscosity is measured by the shear rate for swallowing [10]. The rheological parameters, such as viscosity, elasticity, etc., play vital roles in swallowing. So far, most studies on thickened liquids for patients with swallowing disorders have only focused on the effects of pH, time, and temperature on the rheological behavior of structurally modified foods [11,12] and the viscoelastic properties of the modified foods [13]. However, very few studies on the formulation design and performance of thickening components have been reported.

Konjac glucomannan (KGM), a nonionic water-soluble dietary fiber extracted from the tubers of Konjac, is a highly viscous and thickening agent due to its high molecular mass, with a viscosity much higher than other thickeners at the same concentration, such as carrageenan (KC) and gum arabic. A previous study has confirmed the differences in the viscoelastic and mechanical properties of different gels by evaluating 10 different thickeners. The results also revealed that konjac gum has a stronger viscosity [14]. Additionally, KGM showed significant synergistic thickening effects when combined with xanthan gum (XG), KC, and guar gum (GG) [5,15,16,17,18,19]. KGM is commonly used as a thickening agent in food products. Another study has revealed that the addition of KGM improved the organization and viscosity of the dough [20]. Furthermore, the addition of KGM to starch paste increased its viscosity more than the starch paste without KGM, and the interaction between the KGM and starch inhibited granule association, leading to a significant increase in its viscosity [21].

As a polysaccharide produced from starch by partial enzymatic hydrolysis, maltodextrin is composed of oligomers and/or polymers of α-(1,4) linked DD-glucose [22]. The degree of hydrolysis of maltodextrin is indicated by dextrose equivalent (DE) value [23]. Additionally, maltodextrins with different DE values have different physicochemical properties, including solubility, freezing temperature, viscosity, etc. [24]. As an ingredient in food, maltodextrin offers consistency, viscosity, mild texture, and stability [25]. Meanwhile, recent studies have also shown that maltodextrin and gelatin cross-linking can obtain a stable hydrogel [26].

In this study, the rheological and thickening properties of KGM were investigated, and the optimal formulation of the thickening components for special medical purposes were prepared using KGM as the raw materials. This dysphagia food has a good thickening effect and can meet the needs of patients with different levels of swallowing disorders, and there is no time limit for consumption. This study provides insights into the high value utilization of KGM as a dietary nutritional support for patients with swallowing disorders.

## 2. Results and Discussion

### 2.1. Performance Studies of KGM

#### 2.1.1. Rheological Measurements

According to the experimental results of KGM swelling equilibrium in the pre-experiment, the concentrations of 2, 4, and 6 mg/mL were selected to investigate the properties of KGM. As depicted in Figure 2a, the viscosity of the KGM solution gradually decreased with the increase in shear rate, showing the pseudo-plasticity of shear thinning. At the same shear rate, the viscosity of the solution increased rapidly with the increase in the solution concentration, indicating that the greater the viscosity, the faster it decreases. The viscosity changed slightly, and the shear thinning was not obvious when the concentration of the KGM solution was 2.0 mg/mL. This might be due to the fact that at low concentration, KGM molecules exist in the aqueous solution as trackless wire clusters and that the KGM solution could produce new intermolecular entanglements that are stable enough to hold the network structure of the system upon shearing by low shear rate [27]. When the concentration of the KGM solution was 4.0 mg/mL and 6.0 mg/mL, the shear viscosity decreased rapidly with the increase in shear rate, and the shear thinning characteristic was significant. It is well known that the KGM molecule contains glucose and mannose on which there are rich hydroxyls. The hydroxyl would form hydrogen bonding in KGM solution. There were more average electrostatic attractions (hydrogen bonding) in the molecules, which resulted in viscosity increase [28]. However, the KGM molecular chains stretch along the direction of shear force under the action of shear force [29], the hydrogen bonds are destroyed, and the viscosity of the solution decreases to show the characteristics of the pseudoplastic fluid. Moreover, shear-thinning behavior provides advantageous characteristics such as ease of formation during processing and improved food texture [30], which is conducive to the preparation of dysphagia liquid. Meanwhile, the shear behavior in vitro simulates oral chewing, and the shear-thinning behavior of KGM helps patients to complete swallowing.

As depicted in Figure 2b, the shear stress of KGM solution gradually increased with the increase in shear rate, indicating that the greater the solution concentration, the greater the shear stress [29]. When the KGM solution concentration was 2.0 mg/mL, the shear stress was basically constant and tended to be straight with the change in shear rate. When the KGM solution concentration was 4.0 mg/mL and 6.0 mg/mL, the shear stress increased rapidly with the increase in shear rate. This indicates that KGM has the characteristics of a pseudoplastic fluid.

The viscosity of the KGM solution, shear stress, and shear rate showed an observable nonlinear variation law. The rheological curve of the KGM sol was in accordance with the equation $\sigma =K\gamma^{n}$ [31], where σ is the shear stress; γ is the shear rate; K is the consistency, characterizing the viscosity of the liquid—the larger the K value, the higher the viscosity of the liquid; and *n* is the power law index. The *n* value acts as an indicator of the shear dependence of the viscosity, and *n* < 1 indicates pseudoplastic.

The viscosity change curves of 6.0 mg/mL KGM solution at different temperatures are depicted in Figure 3. Under a certain shear rate, the viscosity of the KGM solution decreased with the increase in temperature. At 20 °C, the viscosity of the solution was 3.4 Pa·s. Similarly, when the temperature was increased to 90 °C, the viscosity of the solution decreased to 0.7 Pa·s. This might have happened due to the increase in temperature, which strengthened the movement of KGM molecules, disrupted the internal structure of the molecules, and reduced interaction force and flow resistance. Furthermore, the results show that the increase in temperature did not completely eliminate the hydrogen bonding, but its force was weakened [32] so that the KGM molecules broke away from the hydrogen bonds with the water molecules, resulting in a decrease in viscosity.

As depicted in Figure 4a, the complex viscosity was characterized by the viscoelastic quality of the polymer fluid, indicating that higher the compound viscosity, the more stable the polymer. The complex viscosity of the KGM solution gradually decreased with the increase in oscillation frequency, confirming the characteristics of a stable structure with shear thinning. Figure 4b illustrates the frequency scan results of KGM with different concentrations under the condition of 1% strain. Moreover, G′ of KGM was similar to G″ at low frequency, and with the gradual increase in frequency, both G′ and G″ of the KGM solution increased, but the increase in G′ was larger than G″. Before the intersection point, i.e., in the low-frequency region, G′ < G″, the system was mainly viscous; after the intersection point, i.e., in the high-frequency region, G′ > G″, the system was mainly elastic, and the modulus of the intersection point increased gradually with the increase in the solution concentration. This might be because, in the low frequency region, the relaxation time is long enough, the deformation occurs slowly, most of the energy is cancelled due to the viscous flow, and the molecules are in a lower energy state. On the contrary, in the high frequency region, the effective relaxation time decreases due to the lack of time for slip between the molecular chains, and the entanglement point resembles a fixed network point, leading to a gradual increase in energy stored in the temporary network structure, and the solution tends to be elastomeric [33].

#### 2.1.2. Swelling Property

As depicted in Figure 5, the swelling degree and swelling volume of KGM increased continuously with the extension of the placement time. In 0–10 min, the swelling degree increased faster, which was due to the oscillation effect that made the KGM molecules evenly dispersed in water and fully interact with water molecules. One more reason could be its good water absorption capacity, which could fully combine a large amount of zwater and accelerate the swelling ratio [34]. In 160–200 min, the swelling time was more sufficient, and the swelling degree increased significantly, reaching equilibrium after 200 min. The backbone conformation of the KGM chain is a two-fold helix stabilized by intra-molecular O-3–O-5′ hydrogen bonds [35], and there is about one ester-bonded acetyl group in every 19 sugar residues in the KGM main chain [36]. Acetyl substituents produce steric effects of acetyl groups to decrease interchain association and improve entropic penalty of chain association, resulting in the high water solubility of KGM [37]. Meanwhile, since KGM is rich in hydroxyl groups, its water absorption could reach 90 times its own mass, suggesting that it is a good thickening effect.

### 2.2. Formulation Optimization of Thickening Component Based on KGM for Special Medical Use

#### 2.2.1. Thickening Properties of Thickeners

As depicted in Figure 6, the viscosity of the KGM solution was the largest with the peak viscosity up to 523 mPa·s, followed by XG solution with the peak viscosity of 304 mPa·s, and GG solution, while LBG solution and KC solution were less viscous with the peak viscosity below 200 mPa·s. The viscosity of all five thickener solutions decreased with the increase in temperature and vice versa. The viscosity of the KGM solution significantly changed with temperature due to the large pseudoplasticity. Similarly, the viscosity decreased by 278 mPa·s with high temperatures. Therefore, it was inferred that the viscosity could return to the initial level under decreased temperature, and LBG and KC solution have less effect on temperature change due to the lower viscosity.

As depicted in Figure 7a, the viscosity of the solution reached the maximum at 525 mPa·s when the ratio of KGM to guar gum was 7:3. Since GG could be fully swollen in cold water, the peak time was very fast, basically 70 s. The viscosity of the solution gradually decreased with the increase in temperature from 50 °C to 95 °C, and gradually increased during the cooling process, but its final viscosity did not exceed the peak viscosity. This is because high temperature will degrade GG [38] and reduce the viscosity of the solution, which is conducive to KGM and GG molecules’ interaction, making the viscosity increase after cooling.

As depicted in Figure 7b, when the ratio of KGM to XG was 7:3, the peak viscosity of the solution was the largest, the peak time was 150 s, and the viscosity reached 920 mPa·s, indicating XG to be an efficient thickener with the characteristics of low concentration and high viscosity. When the percentage of XG was larger, the changes in the viscosity were less with temperature, and the peak time was more than 300 s. This might be due to the thermal stability of XG and to its solution viscosity not changing significantly with the change in temperature. When the ratio of KGM to XG was 3:7, the curve did not have the universality of non-Newtonian fluid, which might be due to the larger XG content, strong hydrophilic nature, and rapid dissolution in water, allowing the KGM and XG component solution final value viscosity to reach more than 1000 mPa·s. When the temperature was high, it was in a flowable manner, and the gel formed after cooling to room temperature, which might be due to a higher temperature, weakened hydrogen bonding between the molecules, or fluidity of the sol. Similarly, when the temperature decreased, the hydrogen bonding between the hydroxyl groups in the polymer chains of the network allowed the polymer chains to entwine with each other.

As depicted in Figure 7c, the viscosity of the solution increased with the increase in the KGM and reached the maximum viscosity of 322 mPa·s when the ratio of KGM to LBG was 7:3. When the temperature was lowered from 95 °C to 50 °C, the viscosity of the solution increased continuously, and its final viscosity could be greater than the peak viscosity. This might be due to the fact that LBG could only be partially dissolved in cold water and fully hydrated when heated to 80 °C [39].

As depicted in Figure 7d, the viscosity of the solution gradually increased with the increase in the ratio of KGM, reaching a maximum viscosity of 244 mPa·s under the ratio of KGM to KC of 7:3. KC could form a transparent and easy flowing solution upon being dissolved in hot water, providing a non-viscous and slippery textured system. When KC was combined with KGM, the viscosity of KC increased compared with the monomeric gum due to the high water-retention and viscosity of KGM.

Compared with the viscosity of the monomeric gum, the viscosity of all four thickeners increased after being combined with KGM, indicating a certain synergistic effect of KGM among all four thickeners. However, the thickening effects of KGM, GG, and XG were significantly improved, while the synergistic effect with LBG and KC was not obvious.

As depicted in Figure 8a, the highest viscosity of KGM, GG, XG, LBG, and KC was found in the compounding ratio of 5:1:2:1:1:1, but it did not meet the consumption requirement of the final product, as it might form a gel after cooling. Polysaccharides could form gel easily after heating and cooling. When the compounding ratio was 6:1:1:1:1, the peak viscosity reached 600 mPa·s, the final viscosity was 933 mPa·s, and the flowability was good after heating and cooling.

As depicted in Figure 8b, the highest viscosity of the compound thickener solution (i.e., 6:1:1:0:0) without LBG and KC was achieved with the addition of KGM, GG, and XG. Similarly, the peak viscosity reached 1068 mPa·s under decreased viscosity, and its final viscosity was only 867 mPa·s; moreover, the compound thickener solution with LBG (i.e., 6:1:1:1:1:0) had the highest viscosity. When it was heated to 95 °C and cooled to room temperature, its viscosity significantly increased, and its final viscosity might reach 1170 mPa·s. This result was in accordance with the aforementioned characteristics of LBG in the compound thickener solution with KC (i.e., 6:1:1:0:1). 

Overall, the KGM, GG, and XG play a significant role in thickening the compounding system. LBG could make the viscosity of the compound thicker and reach the maximum after heating and cooling, while KC could make the surface smoother. Therefore, the compounding ratio of KGM, GG, XG, LBG, and KC was finally determined as 13:2:2:2:1. The viscosity change curve of the compound thickener with temperature is depicted in Figure 8c.

#### 2.2.2. Effect of Maltodextrin on Thickening Component Based on KGM for Special Medical Use

Figure 9 illustrates the peak viscosity, trough viscosity, and final viscosity of the solution in the thickening component based on KGM for special medical use at 10.0 mg/mL. The viscosity of the solution decreased with the gradual decrease in the compound thickener proportion. At the compound thickener concentration of 5.0 mg/mL, the viscosity with different ratios remained constant, with the peak viscosity around 600 mPa·s and the final value viscosity around 900 mPa·s. These results confirmed the role of this compound thickener in a thickening and the lesser effect of maltodextrin on the thickening performance of KGM.

### 2.3. Process Optimization of Thickening Component Based on KGM for Special Medical Use

Figure 10 illustrates the effect of four factors on the preparation of thickening components. With the increase in thickener mass concentration, the sensory score of the prepared KGM thickening components increased and then decreased (Figure 10a), and the highest sensory score was obtained under the thickener mass concentration of 6.0 mg/mL. When the thickener concentration was 4.0 mg/mL, the viscosity of the prepared solution was too low, and the taste was poor. Similarly, when the thickener concentration was too high, the taste of the solution was slightly rough, and the color brightness gradually decreased. When the thickener concentration was 8.0 mg/mL, the viscosity of the solution was too large, and the taste of KGM was too prominent, and the sensory score decreased. Therefore, the thickener mass concentrations of 5.0, 6.0, and 7.0 mg/mL were inferred to meet the requirements of different patients. 

The highest sensory score was achieved under 10.0 mg/mL maltodextrin concentration (Figure 10b), which was the most effective score. At concentrations below 10.0 mg/mL, the brewed solution KGM tasted more pronounced and had a pale yellow color, but the characteristics of maltodextrin were not prominent. The accurate amount of maltodextrin improves the reliability of the product. As a result, when the concentration of maltodextrin was higher than 10.0 mg/mL, the brewed solution was slightly lumpy, and the maltodextrin taste was stronger but could not provide a good taste. Therefore, 10.0 mg/mL was considered as the optimal concentration of maltodextrin. 

The highest sensory evaluation score was obtained at a brewing temperature of 60 °C (Figure 10c). The higher the brewing temperature, the less fluid the product was, and the viscosity of the brewed solution at 90 °C was too thick to maintain a certain shape on the spoon [40]. When the brewing temperature was lower than 60 °C, the viscosity of the solution was thinner, with low color brightness. Therefore, the brewing temperature of 60 °C was considered as the optimum temperature. 

The difference in sensory scores for different placement times was small (Figure 10d), indicating that the set time had a small effect on the sensory aspects of the thickened components. Therefore, a placement time of 10 min after brewing was selected based on the specific habits.

### 2.4. Performance Studies of Thickening Component Based on KGM for Special Medical Use

#### 2.4.1. Rheological Properties

Viscosity reflects the degree of difficulty in destroying the internal structure of the gel. The viscosity of the thickened component gradually decreased with the increase in the shear rate (Figure 11a), showing significant changes. 

In the test model of steady shear, external environment applied a gradually increasing shear force to samples, which influenced the inner structure of system. The shear stress of the thickening component increased with the increase in the shear rate (Figure 11b), illustrating the appearance of a non-linear relationship between shear rate and shear stress, and the system exhibited a pseudoplastic fluid performance [32]. The increasing trend of the thickened component of the three formulations was basically the same, indicating better shear rheological properties of the products.

The effect of temperature on the viscosity of KGM thickening components is depicted in Figure 12. Under the condition of a certain shear rate, the viscosity of the three thickening components gradually decreased with the increase in temperature (Figure 12a). At 20–60 °C, the viscosity decreased rapidly. After 60 °C, the viscosity decreased slowly and gradually stabilized. With the decrease in temperature, the viscosity of the thickening components increased (Figure 12b), showing a significant increase at 60 °C. This might be due to the rebinding of the chemical bonds between the KGM molecules to chains after the decrease in temperature and the entanglement of the polymer chains to increase their viscosity.

As depicted in Figure 13a, the complex viscosity of the thickened component gradually decreased with the increase in oscillation frequency, indicating the characteristics of a shear-thinning non-Newtonian fluid. Figure 13b shows the frequency scan results under the condition of 2% strain. It can also be seen that the loss modulus G″ was lower than the energy storage modulus G′. Meanwhile, as the frequency increased, the G′ and G″ of all samples increased. The energy storage modulus could be temporarily stored in the deformation and later recovered, while the loss modulus is directly converted into shear heat disappearance. tan δ, the ratio of G″ to G′, is another characteristic value for evaluating the viscoelasticity of materials. The tan δ < 1 indicated elastic behavior predominantly, while tan δ > 1 indicated viscous behavior predominantly [41]. With the increase in frequency, the tan δ of the samples decreased and remained in the range of 0.1–1 (Figure 13c), which would be suitable for dysphagia liquid food. The tan δ of sample 1# increased temporarily, which may be due to the weak crosslinking of polysaccharide molecular chains at lower concentration, while the crosslinking degree of polymer decreased, the structure was loose, and the elastic strength increased with frequency. Furthermore, the thickened component system showed a thicker state.

The three KGM thickening components are depicted in Figure 14. The 1# thickening component was nectar-thick level (350 mPa·s) with a thick soup-like consistency and a minimal consistency suitable for elderly people with mild chewing disorders. The 2# thickening component was a honey-thick level (1250 mPa·s) with a yogurt-like consistency and a moderate consistency, suitable for elderly people with moderate chewing disorders or mild swallowing difficulties. The 3# thickening component was a pudding-thick level (1810 mPa·s) with a high consistency and a paste-like consistency, which could improve the dietary safety of aging people with significant swallowing disorders.

#### 2.4.2. Particle Shape and Particle Size Distribution

Figure 15 illustrates the particle shapes and particle sizes of different raw material powders. KGM (a) and KC (e) had a blocky particle shape and particle sizes around 150 μm. The particles of GG (b) were irregularly striped and had depressions on the particle surface. The particles of XG (c) were irregularly shaped blocks with uneven and unequal particle surfaces, with the size of large particles being about 100 μm and small particles about 30 μm. The particles of LBG (d) were long strips with smooth particle surfaces. The particles of maltodextrin (f) were more disordered, with an overall blocky shape and smaller particle size of about 5–10 μm. The small pores perceived on the surface of KC and maltodextrin particles facilitated the wetting and punching of the powder.

The thickening components of three formulations had different particle sizes after mixing due to different particle sizes of the raw materials. It is obvious that the 1# thickening component had more small particles, while the 2# and 3# thickening components had larger granularity and mostly lumpy particles. This might be attributed to the fact that the maltodextrin content in the 1# thickening component was higher. Since maltodextrin posed lumpy and small particles, the particle size of the 1# thickening component was smaller. Meanwhile, the surface of the particles of different formulations of thickening components were not flat, with certain depressions and folds, and some small particles were attached to the depressions of large particles to fill the gap between the particles. However, the small particles could quickly be dissolved during brewing.

The particle size range of the 1# thickening component was 5.9–192 μm, while for the 2# and 3# thickening components, it was about 11.9–272 μm (Figure 16). Moreover, a maximum volume peak located at the minimum particle size was observed in this particle size range. This might be due to the significant effects of the smaller particle size and higher mass fraction of maltodextrin on the particle size distribution of the thickening components. The particle size of 1# was small, with an average of 12.1 μm. However, the particle sizes of 2# and 3# were larger than 1#, with an average of 25.9–26.9 μm. The difference between 2# and 3# was not significant, probably due to the relatively high mass fraction of the compound thickener in 2# and 3#, which reduced the effect of maltodextrin on the particle size distribution of the thickening components.

#### 2.4.3. Crystallization Performance

The XRD patterns of the crystals have sharp peaks, the X diffraction patterns of the non-crystals are broad diffuse shapes, and the amorphous samples are more prone to moisture absorption and dissolution [42]. GG, XG, LBG, and maltodextrin showed no crystalline peak formation, while KGM and KC showed distinct peak shapes, and KGM had three diffraction peaks near 2θ = 28.08°, 40.2°, and 58.5° (Figure 17). XRD plots of the three formulations of the thickening components showed multiple sharp peaks, with the crystallinity of 25.8 for 1#, 30.7 for 2#, and 31.9 for 3#. Compared with the XRD plots of the raw material powders, the positions of the peaks in the thickening components were mainly of the KGM, GG, and maltodextrin peaks, which were consistent with the formulation process. The KC peaks were more intense, but the effect of KC on the peak shape of the thickened component was not significant due to its low content in the thickened component. The 1# thickened component had less peak intensity due to the low compound thickener content and KGM with crystallinity, attributed to good solubility.

#### 2.4.4. FT-IR Analysis

It can be seen from Figure 18 that KGM had characteristic absorption peaks of glycans at specific wavenumbers and that there was an intramolecular or intermolecular hydroxyl (–OH) vibrational absorption peak at 3620 cm^−1^, which had a very wide peak type, indicating the existence of a large number of extensive hydrogen bonds composed of hydroxyl groups in KGM [43]. XG and GG had similar hydroxyl absorption peaks as KGM. The thickening component had a hydroxyl vibration absorption peak at 3355 cm^−1^ and a wide absorption band, indicating the presence of multi-molecular associative hydrogen bonds. Compared with the hydroxyl vibration peak of pure KGM, GG, and XG, the vibration peak was strengthened and shifted to a certain direction of low wave number. This was the result of intermolecular interaction between polysaccharides and showed an enhanced synergistic thickening effect. 

The absorption peak of KGM at 1732 cm^−1^ was the stretching vibration of the carbonyl group (C=O), which was caused by the acetyl group on the glucose unit in the molecular structure of KGM. Although its content was low, it had a significant effect on the crystallinity and water solubility of KGM. The absorption peak near 804 cm^−1^ was the mannose characteristic absorption peak of colloidal polysaccharides [44]. For the interaction between KGM, KC and LBG, it was generally believed that the unsubstituted “smooth” region of mannan chain binds to the carrageenan helix [45]. The spectra of the three thickening components were basically the same. The results show that there was physical mixing between the materials and that the hydrogen bond in the colloidal polysaccharide plays an important role in the synergistic thickening of the complex system.

## 3. Materials and Methods

### 3.1. Materials

KGM was purchased from Yizhi Konjac Biotechnology Co., Ltd. (Yichang, China). XG (Yuanye Bio-Technology Co., Ltd. (Shanghai, China)). LBG, GG, and KC (Wanbang Chemical Technology Co., Ltd. (Zhengzhou, China)), KC is a compound of κ-Carrageenan and potassium chloride. Maltodextrin was derived from corn and acquired from Xiwang Sugar Co., Ltd. (Zouping, China) with a DE of 18 to 20. All these materials were of high food grade. 

### 3.2. Performance Studies of KGM

#### 3.2.1. Rheological Properties

Accurately, 0.10, 0.20, and 0.30 g of KGM were weighed and dissolved in 50 mL of deionized water to prepare the KGM solutions of 2.0, 4.0, and 6.0 mg/mL. The rheological properties of the KGM solutions were characterized using a HAAKE MARS 60 rotational rheometer with a plate measurement system of 35 mm diameter (plate spacing of 1 mm). The temperature of the sample bench was set to 25 °C, and the shear rate was increased from 0.01 s^−1^ to 200 s^−1^ to record the rheological curves of the KGM solutions with different mass concentrations. The shear rate was fixed at 10 s^−1^, and the temperature was increased from 10 °C to 100 °C at a rate of 18 °C/min. The viscosity changes data of 6.0 mg/mL KGM solutions at different temperatures were recorded to study the static rheological properties. The temperature was set at 25 °C, the strain range was 0.01 percent–100 percent, and the frequency was 1.0 Hz. The linear viscoelastic region of the sample was measured, and the strain was experimentally found to be 1%. A frequency scan and a frequency setting of 0.1–100 Hz were performed in the linear viscoelastic region with a fixed temperature of 25 °C to determine the changes in energy storage modulus G′, loss modulus G″, and complex viscosity η′ in the KGM solution with oscillation frequency and to study the dynamic rheological properties of the KGM solutions.

#### 3.2.2. Swelling Characteristics

Accurately, 0.10 g of KGM (*W_d_*) was dispersed in a centrifuge tube containing 10 mL of deionized water, shaken, and allowed to stand for 0, 5, 10, 20, 40, 80, 120, 160, 200, 240 min, centrifuged at 4000 rpm for 10 min, and finally, the supernatant was discarded, and the resultant was weighed (*W_s_*). The swelling ratio was calculated using Formula (1) [46].
*E_sr_* (%) = ((*W_s_* − *W_d_*)/*W_d_*) × 100,(1)
where *E**_sr_* is the water absorption (%wt) of the KGM, and *W_d_* and *W_s_* are the weights (g) of the samples in the dry and swollen states, respectively.

### 3.3. Formulation Optimization of Thickening Component Based on KGM for Special Medical Use

#### 3.3.1. Thickening Properties of Thickeners

Approximately 0.25 g of KGM, GG, XG, LBG, and KC were dispersed in 50 mL of distilled water at 30 °C. The samples were shaken gently for 2 min—to ensure that they were fully swollen—and kept for 60 min. The viscosity of each solution was monitored by an RVA-TM rapid viscosity analyzer. The heating parameters: 50–95 °C, the cooling parameters: 95–50 °C, speed: 960–160 rpm.

A certain mass of KGM and GG was weighed and mixed to the mass ratios of 7:3, 6:4, 5:5, 4:6, and 3:7 by three-dimensional mixing, respectively. Each sample was dispersed in a certain volume of distilled water at 30 °C, shaken gently for 2 min to make them fully swollen, and left for 60 min to prepare a 5.0 mg/mL solution of the compound thickener. The viscosity of each solution was measured by an RVA-TM rapid viscosity analyzer. The temperature rise parameter: 50–95 °C, temperature drop parameter: 95–50 °C, speed: 960–160 rpm. Furthermore, a definite mass of XG, LBG, and KC was weighed and mixed with KGM according to the above ratio and method, preparing a compound thickener solution. Finally, the viscosity of each solution was monitored.

According to the single factor experiment of thickener, the characteristics of each thickener were clarified, and the proportion of compound thickeners was designed on this basis. A certain mass of KGM, GG, XG, LBG, and KC was weighed and mixed to obtain the mass ratios of (1) 6:1:1:1:1; (2) 5:2:1:1:1:1; (3) 5:1:2:1:1; (4) 5:1:1:2:1; (5) 5:1:1:1:2; (6) 2:2:2:2:2; (7) 3:2:2:1:1:2; (8) 6:1:1:0:0; (9) 6:1:1:1:0; (10) 6:1:1:0:1; (11) 13:2:2:2:2:1. The samples were dispersed in a certain volume of distilled water at 30 °C, shaken gently for 2 min to make them fully swollen, and left for 60 min to prepare 5.0 mg/mL of the compound thickener solution. The viscosity of each solution was monitored by an RVA-TM rapid viscosity analyzer. The heating parameters: 50–95 °C, cooling parameters: 95–50 °C, speed: 960–160 rpm.

#### 3.3.2. Effect of Maltodextrin on Thickening Component Based on KGM for Special Medical Use 

A certain mass of the compound thickener (KGM/GG/XG/LBG/KC = 13:2:2:2:1) and maltodextrin with the mass ratio of the composite thickener and maltodextrin (10:0, 7:3, 6:4, 5:5, 4:6, 3:7, 0:10) were dispersed in a certain volume of distilled water at 30 °C, shaken gently for 2 min to make them fully swollen, and left for 60 min to prepare 10.0 mg/mL of the thickening components for special medical use. The viscosity of each solution was monitored by an RVA-TM rapid viscosity analyzer. The heating parameters: 50–95 °C, cooling parameters: 95–50 °C, speed: 960–160 rpm. 

Approximately 0.25 g of the compound thickener (KGM/GG/XG/LBG/KC = 13:2:2:2:1) and a certain mass of maltodextrin were mixed to obtain the mass ratios of compound thickener to maltodextrin of 1:3, 1:2, 1:1, 2:1, 3:1. Later, each sample was dispersed in 50 mL of distilled water at 30 °C, shaken gently for 2 min to make them fully swollen, and left for 60 min to prepare the KGM-based thickening components. The viscosity of each solution was monitored by an RVA-TM rapid viscosity analyzer. The heating parameters: 50–95 °C, cooling parameters: 95–50 °C, speed: 960–160 rpm.

### 3.4. Process Optimization of Thickening Component Based on KGM for Special Medical Use

The samples were dispersed in 150 mL of distilled water, shaken gently for 2 min to make them fully swollen, and left for a certain period of time to prepare the KGM thickening components with the mass concentrations of 4.0, 5.0, 6.0, 7.0, and 8.0 mg/mL of the compound thickening agent, respectively. The samples were evaluated by a panel of ten semi-trained judges comprising five males and females tested using sensory evaluation standard (Table 1) [47], on the basis of color, scent, taste, dissolvability, and structural state. To avoid sensory fatigue, the sensory test was divided into four separate sessions. Each panelist was requested to assess only one attribute in a session [48]. The following factors were used as indicators for investigating the effects of the sensory scores on the thickening components: (1) mass concentration of the thickener: 4.0, 5.0, 6.0, 7.0, 8.0 mg/mL; (2) mass concentration of maltodextrin: 5.0, 10.0, 15.0, 20.0, 25.0 mg/mL; (3) brewing temperature: 30, 45, 60, 75, 90 °C; (4) left for 10, 20, 30, 40, 50 min.

### 3.5. Performance of Thickening Component Based on KGM for Special Medical Use

According to Table 2, a certain amount of the compound thickener (KGM/GG/XG/LBG/KC = 13:2:2:2:1) and 0.50 g of maltodextrin were weighed, mixed and prepared as 1#, 2#, 3# of the KGM thickening components for special medical purposes.

#### 3.5.1. Rheological Properties

The 1#, 2#, 3# of the KGM thickening components were dispersed in 50 mL of distilled water at 60 °C, shaken gently for 2 min to make them fully swollen, and ultrasonically treated for 10 min. The rheological parameters were monitored using a HAAKE MARS 60 rotational rheometer with a 35 mm diameter (plate spacing of 1 mm) plate measuring system. The sample bench temperature was set to 25 °C, and the shear rate was increased from 0.01 s^−1^ to 200 s^−1^ to investigate the relationship between the shear rates of the thickening components, viscosity, and shear force. The shear rate was fixed at 10 s^−1^, and the temperature was increased from 20 °C to 100 °C and then decreased from 100 °C to 20 °C with a variable temperature rate of 0.27 °C/s to investigate the effect of temperature on the viscosity of the thickened components. The linear viscoelastic zone of each sample was measured at 25 °C, with the strain range varying from 0.01% to 100%, and the frequency at 1.0 Hz to determine the optimal strain value. A frequency scan was performed in the linear viscoelastic zone with a fixed temperature of 25 °C and a frequency setting of 0.1–100 Hz to determine the variation in energy storage modulus G′, loss modulus G″, and complex viscosity η′ of the thickened components with oscillation frequency. Finally, the dynamic rheological properties were determined.

#### 3.5.2. Particle Shape and Particle Size Distribution Analysis

Appropriately, 1#, 2#, and 3# of KGM thickening components were taken, fixed on the sample table with conductive adhesive, and sprayed with gold. The microstructure of each sample was observed under FEI Verios 460 scanning electron microscope. Appropriately, 1#, 2#, 3# of the KGM thickening components were taken, and the particle size of each sample was determined by an S3500 McKick laser particle size analyzer.

#### 3.5.3. Crystallization Performance 

Appropriately, 1#, 2#, and 3# of KGM thickening samples were taken, and the crystallinity of each sample was measured by a Smart Lab 9 kW X-ray diffractometer. The measurement range was 5°−80°, the step size was 2θ (0.02°), and the velocity was 10°/min.

#### 3.5.4. FT-IR Analysis

Appropriately, 1#, 2#, and 3# of KGM thickening samples were taken, and mixed with KBr (less than 200 mesh, spectrally pure, dried at 130 °C for more than 4 h and then placed in a dryer) at mass ratio of 1:100. FT-IR spectroscopy was obtained using a Vector-22 FT-IR spectrometer (Bruker, Germany) in the range of 4000–400 cm^−1^ [49].

### 3.6. Data Processing

The experimental results are expressed as mean ± SEM (standard error of the mean) of the triplicate measurements. The statistical analysis was performed using Origin 8.0 software.

## 4. Conclusions

In this study, the rheological and thickening properties of KGM were characterized. The results indicate that KGM had good thickening and swelling properties. The greater the concentration, the better the thickening and swelling properties. The KGM solution exhibited the characteristics of shear thinning and non-Newtonian fluid. The viscosity, shear stress, and shear rate of the solution exhibited significant nonlinear variation when the ratio of KGM, GG, XG, LBG, and KC was 13:2:2:2:1. The thickening performance of the composite thickener showed the best results. The optimal preparation process of the KGM thickener was as follows: the concentration of the compound thickener was 5.0–7.0 mg/mL, the concentration of maltodextrin was 10.0 mg/mL, the brewing temperature of the thickening components was 60 °C, with no restriction on consumption time. The three thickeners exhibited three different thickening effects of nectar-thick level, honey-thick level, and pudding-thick level suitable for the requirements of people with different degrees of chewing disorders. Three kinds of special medical thickening components based on KGM exhibited pseudoplastic fluid characteristics, and the viscosity decreased rapidly with the increase in shear rate. The powder particle diameter of the thickening component was small, and the average particle size was 12.1–26.9 μm. The particle size was relatively concentrated. The crystallinity was small, between 25.8–31.9 μm, and the dissolution performance was good. This study provides a theoretical basis and technical reference for dietary nutritional support of patients with dysphagia.

## Figures and Tables

**Figure 1 molecules-27-02194-f001:**
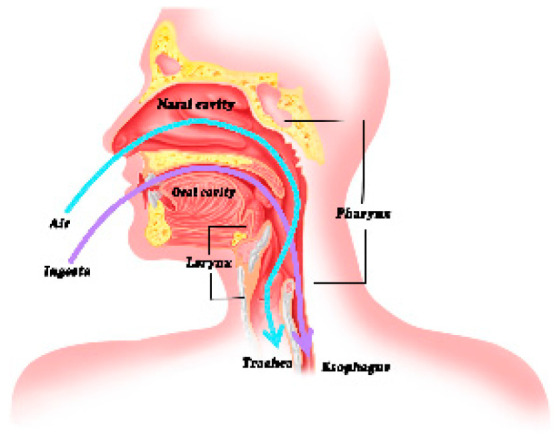
Swallowing pattern diagram.

**Figure 2 molecules-27-02194-f002:**
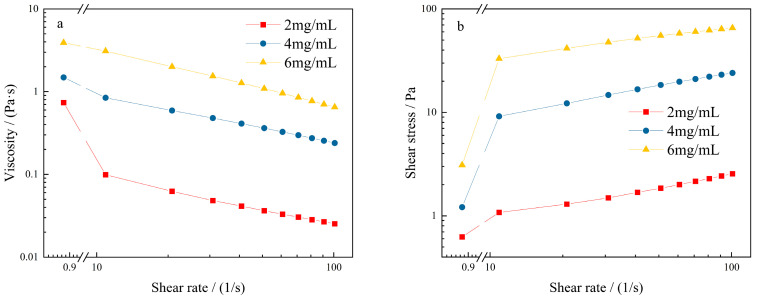
Rheological curves of KGM with different concentrations: (**a**) relationship between shear rate and viscosity; (**b**) relationship between shear rate and shear stress.

**Figure 3 molecules-27-02194-f003:**
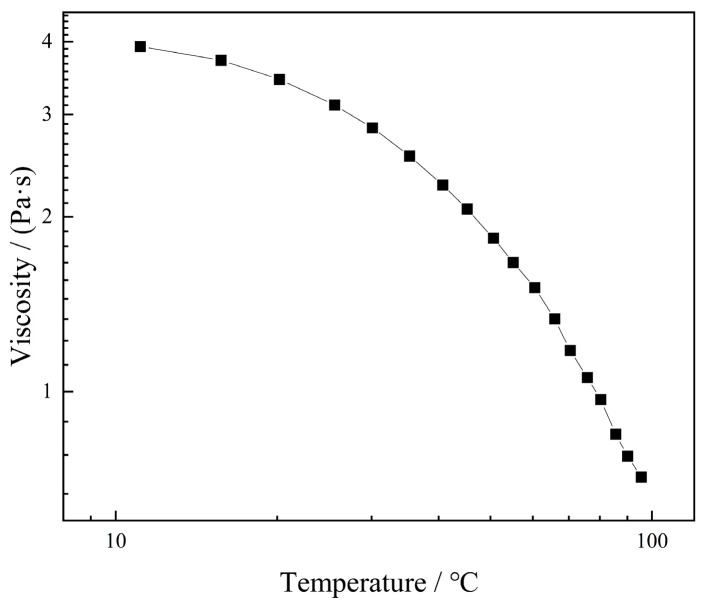
Effect of temperature on viscosity of KGM.

**Figure 4 molecules-27-02194-f004:**
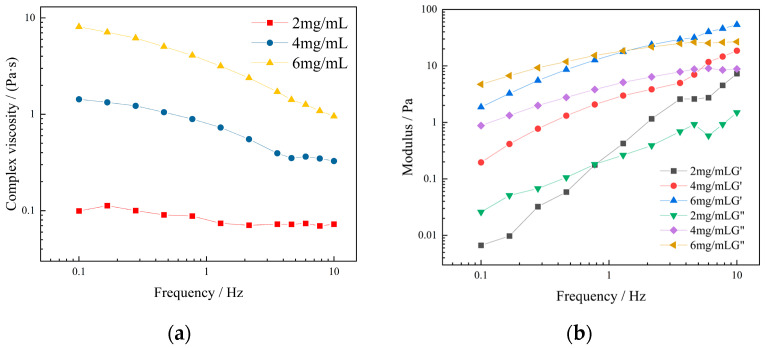
Dynamic viscoelastic curves of KGM with different concentrations: (**a**) relationship between frequency and viscosity; (**b**) relationship between frequency and modulus.

**Figure 5 molecules-27-02194-f005:**
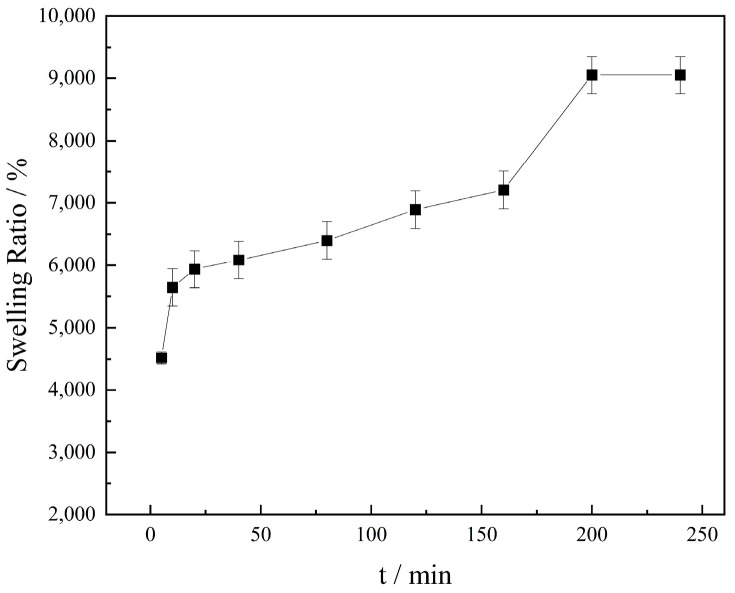
Swelling property of KGM.

**Figure 6 molecules-27-02194-f006:**
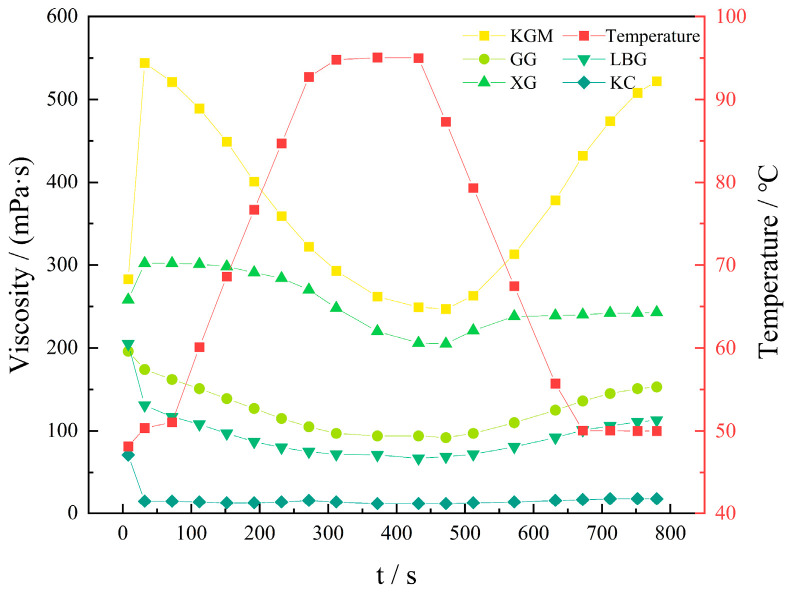
Viscosity curves of 5 kinds of thickeners.

**Figure 7 molecules-27-02194-f007:**
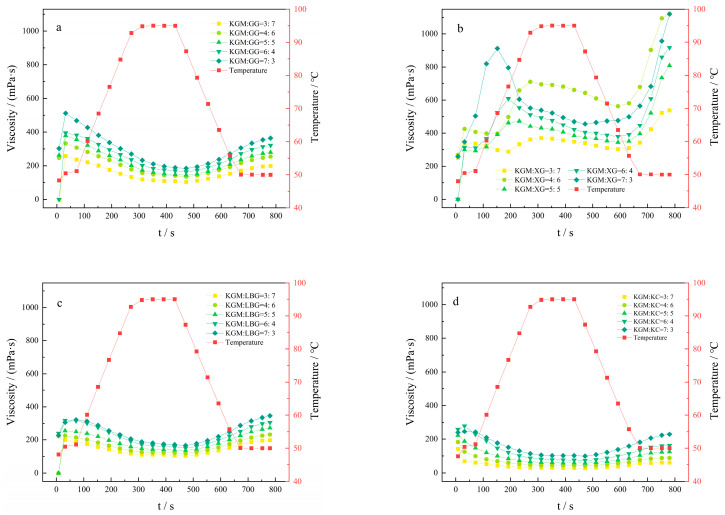
Viscosity curves of KGM and single thickener: (**a**) KGM mixed with GG; (**b**) KGM mixed with XG; (**c**) KGM mixed with LBG; (**d**) KGM mixed with KC.

**Figure 8 molecules-27-02194-f008:**
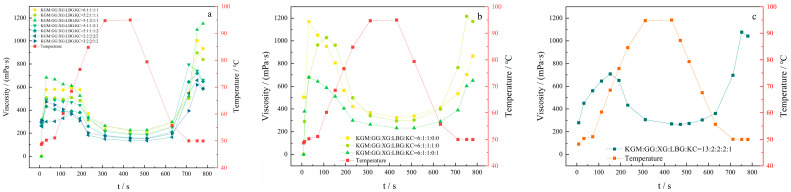
Viscosity curves of 5 thickeners mixed in different proportions: (**a**) the compound of KGM, GG, XG, LBG and KC; (**b**) the compound of KGM, GG, and XG; (**c**) the compound thickener.

**Figure 9 molecules-27-02194-f009:**
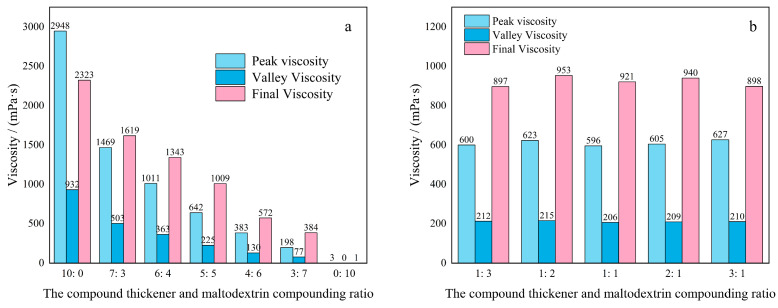
Viscosity changes in compound thickeners and maltodextrin in different proportions: (**a**) the total concentration of mixed solution at 10.0 mg/mL; (**b**) the concentration of compound thickener at 5.0 mg/mL.

**Figure 10 molecules-27-02194-f010:**
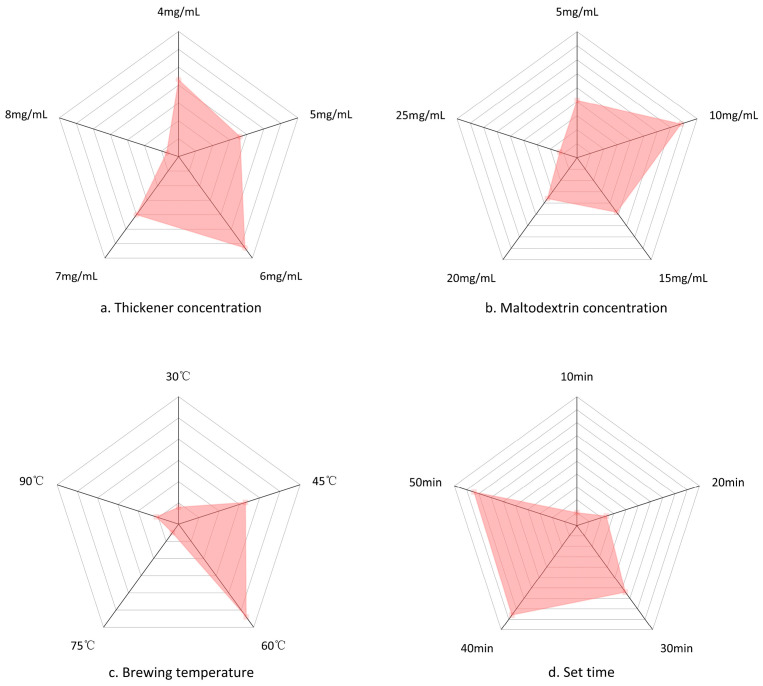
Effect of four factors on sensory scores of thickening components: (**a**) thickener concentration; (**b**) maltodextrin concentration; (**c**) brewing temperature; (**d**) set time.

**Figure 11 molecules-27-02194-f011:**
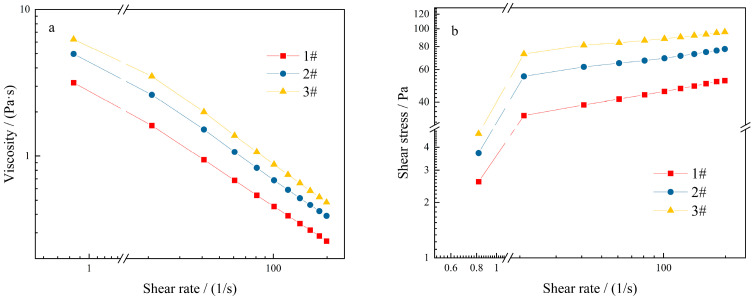
Rheological curves of thickening components with different formulations: (**a**) relationship between shear rate and viscosity; (**b**) relationship between shear rate and shear stress.

**Figure 12 molecules-27-02194-f012:**
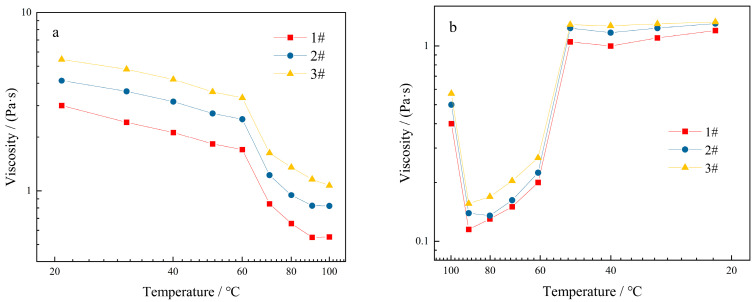
Effect of temperature on viscosity of thickening components with different formulations: (**a**) temperature rise; (**b**) temperature drop.

**Figure 13 molecules-27-02194-f013:**
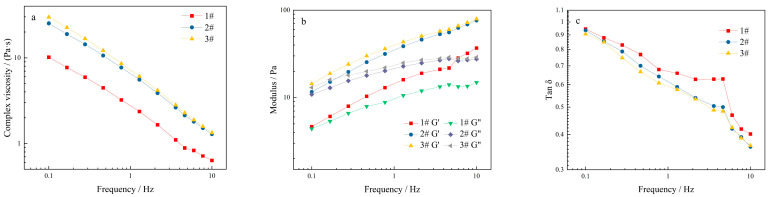
Dynamic viscoelastic curves of thickening components with different formulations: (**a**) relationship between frequency and viscosity; (**b**) relationship between frequency and modulus; (**c**) relationship between frequency and tan δ.

**Figure 14 molecules-27-02194-f014:**
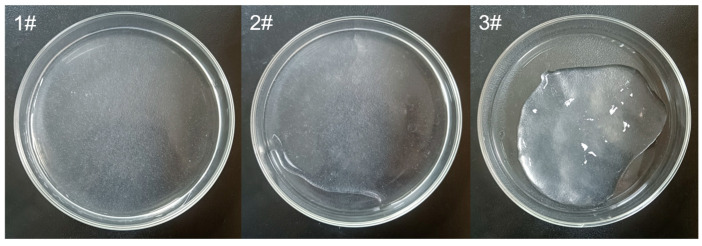
Thickened liquids for patients with KGM-mediated dysphagia.

**Figure 15 molecules-27-02194-f015:**
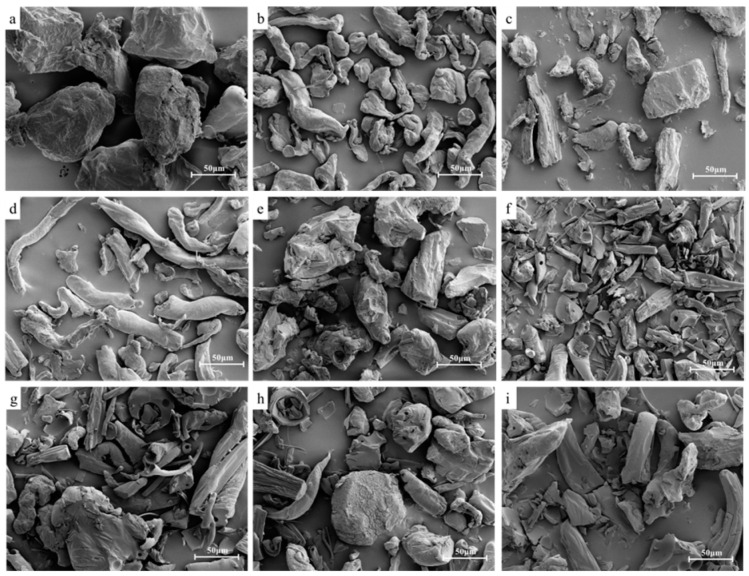
SEM image of thickeners and thickening components based on KGM for special medical use: (**a**) KGM; (**b**) GG; (**c**) XG; (**d**) LBG; (**e**) KC; (**f**) maltodextrin; (**g**) 1# thickening component; (**h**) 2# thickening component; (**i**) 3# thickening component.

**Figure 16 molecules-27-02194-f016:**
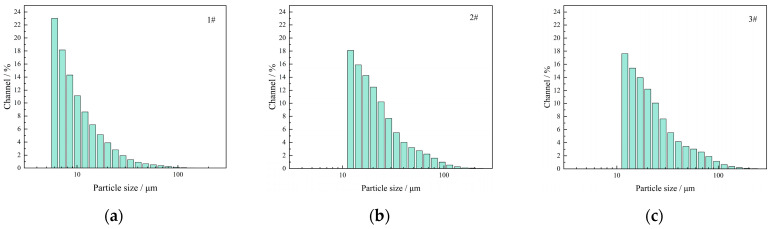
Particle size distribution of thickening components with different formulations: (**a**) 1# thickening component; (**b**) 2# thickening component; (**c**) 3# thickening component.

**Figure 17 molecules-27-02194-f017:**
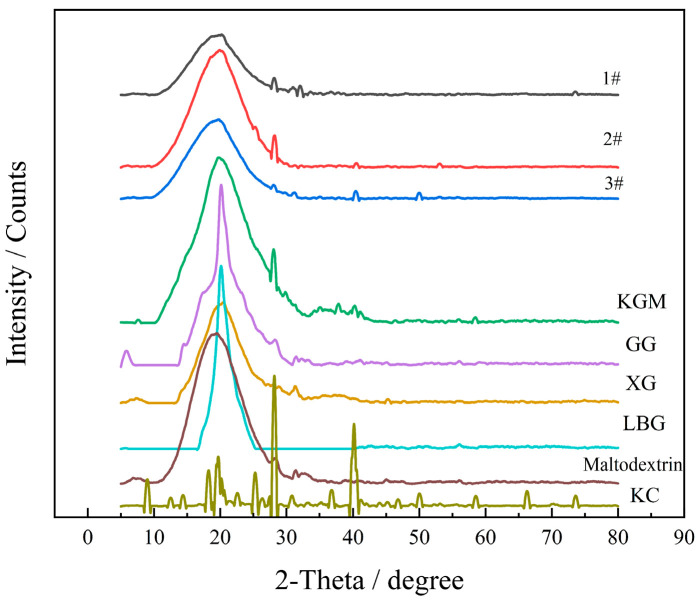
XRD patterns of thickeners and thickening components based on KGM for special medical use.

**Figure 18 molecules-27-02194-f018:**
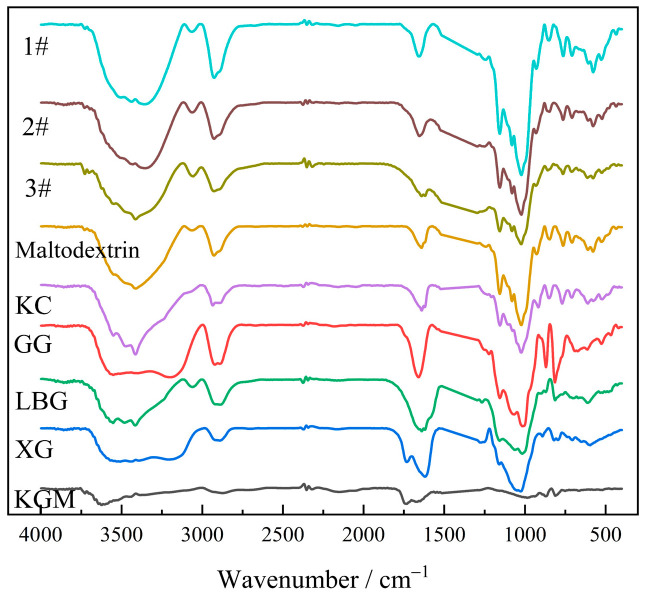
FT-IR patterns of thickeners and thickening components based on KGM for special medical use.

**Table 1 molecules-27-02194-t001:** Standard for sensory evaluation.

Sensory Indicators	14–20 Points	7–13 Points	6 Points
Color (20 points)	Aggregate score	Normal color, no gloss	Uneven, impure color
Scent (20 points)	Unique smell of KGM	No strange smell	Other peculiar smell
Taste (20 points)	Delicate and smooth taste	Grainy taste	Rough taste with dry powder
Dissolvability (20 points)	Dissolve quickly and evenly	Slightly agglomerated, dissolve after stirring	More agglomerates, not easy to dissolve
Structural state (20 points)	Suitable viscosity without impurities	High viscosity with a small amount of impurities	Low viscosity with obvious impurities
Aggregate score	100 points (full score)		

**Table 2 molecules-27-02194-t002:** Content of each component of the 3 formulations of thickening component based on KGM for special medical use.

Number	1#	2#	3#
Mass fraction of compound thickener/%	33.3	37.5	41.2
Mass fraction of maltodextrin/%	66.7	62.5	58.8

## Data Availability

All the data generated during this study are included in this article.

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
