# Peer review of "Preparation and Performance of Thickened Liquids for Patients with Konjac Glucomannan-Mediated Dysphagia"

_molecules, 2022, doi:10.3390/molecules27072194_

Round 1

Reviewer 1 Report

-Is Figure 1 a drawing of the Authors made for this paper? If not, permissions should be asked for and added in the legend.

-L68-72: More precise infomration is needed on the aims, as to highlight the innnovative aspects of this work. This is very important, given the substantial volume of works on similar systems.

-Methods: The texts needs to provide an exmlanation of the choosing of the studied concentrations.

-More details on the sensory panel composition and training are needed.

-2.3.4: A more detailed discussion is needed on the solid nanostructure, as well in linkiing teh properties of the solid to those of the liquid material.

-

Author Response

Dear Editors and Reviewers,

     Thank you for your letter and comments concerning our manuscript entitled “Preparation and Performance of Thickened Liquids for Patients with Konjac Glucomannan Mediated Dysphagia” (ID: molecules-1598956). Those comments are all valuable and helpful for revising and improving our paper. We have studied comments carefully and have made correction which we hope meet approval. Revised portion were marked in red in the paper. The main correction in the paper and the response to the reviewers’ comments are as following:

  1. Is Figure 1 a drawing of the Authors made for this paper? If not, permissions should be asked for and added in the legend.

Response: Yes, Figure 1 was created by us specifically for this paper.

  1. L68-72: More precise infomration is needed on the aims, as to highlight the innnovative aspects of this work. This is very important, given the substantial volume of works on similar systems.

Response: Thank you for your advice. We have added the innovative aspects in the manuscript, and the revised part has been marked in red. (L78-80)

  1. Methods: The texts needs to provide an exmlanation of the choosing of the studied concentrations.

Response: In the previous stage, the mass concentration range of KGM was preliminarily determined to be 0‑10 mg/mL through the literature. Based on this, we conducted pre-experiments to measure the viscosity of KGM solutions with different concentrations. The results showed that when the addition of KGM increased to a certain extent, the swelling equilibrium was achieved, and the viscosity increased was not as obvious as before. Thus, we chose 2, 4, 6 mg/mL as the studied concentrations. We have added an explanation in L86-88 of the manuscript.

  1. More details on the sensory panel composition and training are needed.

Response: Thank you for your advice. We have supplemented the sensory evaluation and marked it in red in L497-501 of the manuscript.

  1. 2.3.4: A more detailed discussion is needed on the solid nanostructure, as well in linkiing teh properties of the solid to those of the liquid material.

Response: We have added a description of KGM solid structure, chemical groups and solubility in the manuscript at L163-168.

    Thank you very much for your comments and suggestions.

Looking forward to hearing from you.

Thank you and best regards.

Yours sincerely,

Wen Zhang

March. 19, 2022

Reviewer 2 Report

The introduction to the subject of research and the validity of undertaking them are quite well done. 

3.1 materials - it is weak, the exact same sentence is read with a subtle change of the raw material and the manufacturer. Signs to write one sentence of the introduction and then point out the raw materials and in the producer's parentheses 

line 384, while the shear rate is expressed in seconds, the temperature rise rate is rather in C / min. Are the sample systems just heated? whether it was also cooled ... it does not result from the methodology 

line 402-404 .. the authors write about calculating the solubility from the formula and in fact they calculate the swelling 

line 401 and 426, please standardize the units of turnover r/min or RPM

lina 498 is not a conclusion but an indication of the methods used in the work. 

figure 7 in the legend should be the word temperatuta and not just an abbreviation 

Figure 8 and 9 and 13 axes describing the viscosity should have sme max values 

Author Response

Dear Editors and Reviewers,

     Thank you for your letter and comments concerning our manuscript entitled “Preparation and Performance of Thickened Liquids for Patients with Konjac Glucomannan Mediated Dysphagia” (ID: molecules-1598956). Those comments are all valuable and helpful for revising and improving our paper. We have studied comments carefully and have made correction which we hope meet approval. Revised portion were marked in red in the paper. The main correction in the paper and the response to the reviewers’ comments are as following:

  1. 3.1 materials - it is weak, the exact same sentence is read with a subtle change of the raw material and the manufacturer. Signs to write one sentence of the introduction and then point out the raw materials and in the producer's parentheses

Response: The modification has been made according to your suggestion, and the revised part has been marked in red in 3.1 material. (L420-424)

  1. line 384, while the shear rate is expressed in seconds, the temperature rise rate is rather in C/min. Are the sample systems just heated? whether it was also cooled ... it does not result from the methodology

Response: The unit of temperature rise rate has been changed to ℃/min. The KGM solution was heated to show the effect of temperature increase on the apparent viscosity of the solution, as shown in Figure 3. The subsequent method has included the viscosity change of the KGM solution after cooling. (L435)

  1. line 402-404. the authors write about calculating the solubility from the formula and in fact they calculate the swelling

Response: We are very sorry for the mistake, and we have corrected the manuscript with red markings and cited the method literature. The corresponding Figure 5 has also been changed. (L445-450)

  1. line 401 and 426, please standardize the units of turnover r/min or RPM

Response: The units has been changed to RPM, and the revised portion was marked in red in the manuscript. (L447)

  1. lina 498 is not a conclusion but an indication of the methods used in the work.

Response: Thank you for your suggestion, and the original section of Line 551 has been deleted.

  1. figure 7 in the legend should be the word temperatuta and not just an abbreviation.

Response: These recommendations have been changed in Figure 6, and the same changes are shown in Figures 7 and 8 (the figure number has been changed, this is the latest number).

  1. Figure 8 and 9 and 13 axes describing the viscosity should have sme max values.

Response: Thank you for your advice. Figure 7 and 8 and 12 (the figure number has been changed, this is the latest number) have been modified.

    Thank you very much for your comments and suggestions.

Looking forward to hearing from you.

Thank you and best regards.

Yours sincerely,

Wen Zhang

March. 19, 2022

Reviewer 3 Report

The manuscript present the interesting topic about the development of KGM based thickener. I would like to give some suggestions as below:

  1. Please give the reference of figure 1.
  2. Line 26: Please give the information about the using of maltodextrin for studying in this work.
  3. Lines 81-84: Please give the reference for the sentence “This might be due to…by the low shear rate”.
  4. Lines 87-91: Please explain more about the relation between hydrogen bond and viscosity as well as rheology behavior.
  5. Line 98: please discuss about the relation between pseudoplastic behavior of KGM and the rheology behavior requirement of dysphagia liquid food.
  6. Lines 112-115: Please explain more how to know that the increase of temperature did not eliminate the hydrogen bonding between KGM and water.
  7. Figure 4: Please discuss about tan d (G”/G’) of samples to confirm the gel like behavior. Tan d in the range of 0.1-1 is suitable for dysphagia liquid food.
  8. Line 137: Could you please give the information of melting point temperature of KGM?
  9. Line 176: please give the reference showed that high temperature degrade GG.
  10. Why was pure KGM not suitable for using as thickener in dysphagia food? Why was the thickener formula needed to develop?  
  11. Line 228: According to Figure 8c, the case of LBC showed low viscosity compared to other cases. Please give the reason for the sentence “LBG could make the viscosity of the compound thickener…”.
  12. Lines 305-312: Which shear rate was used to monitor the viscosity and consistency type of samples? Which shear rate is conventionally used to monitor the viscosity of foods during swallowing?  
  13. Please show the FTIR results for studied thickener and thickening component to confirm synergistic of components.
  14. Line 415: Please give the detail of shaking speed and time for making fully swollen.
  15. Lines 423-425: please give the information of design of experiment (DOE) method.
  16. Line 432: Why was the compound thickener of KGM:GG:XG:LBG:KC = 13:2:2:2:1 chosen for studying the effect of maltodextrin?
  17. Issues 3.3 and 3.4: What is the mixing method of component of thickener? Is it the physical mixing?
  18. Table 2: Could you please give the information of the range of experimental condition design?

Author Response

Dear Editors and Reviewers,

     Thank you for your letter and comments concerning our manuscript entitled “Preparation and Performance of Thickened Liquids for Patients with Konjac Glucomannan Mediated Dysphagia” (ID: molecules-1598956). Those comments are all valuable and helpful for revising and improving our paper. We have studied comments carefully and have made correction which we hope meet approval. Revised portion were marked in red in the paper. The main correction in the paper and the response to the reviewers’ comments are as following:

  1. Please give the reference of figure 1.

Response: Figure 1 was created by us specifically for this paper.

  1. Line 26: Please give the information about the using of maltodextrin for studying in this work.

Response: An introduction to maltodextrins has been added, and marked in red on L68-75 in the manuscript.

  1. Lines 81-84: Please give the reference for the sentence “This might be due to…by the low shear rate”.

Response: The reference is as follow: Elisabeth K. Hill, Yalin Wei, Dave E. Dunstan. Direct measurement of polymer segment orientation and distortion in shear: Semi‑dilute solution behavior of a conjugated system. AIP Conference Proceedings. 2004, 708(1), 209-212. [https://doi.org/10.1063/1.1764117]. And it has been added to the manuscript. (L96)

  1. Lines 87-91: Please explain more about the relation between hydrogen bond and viscosity as well as rheology behavior.

Response: Thank you for your advice. More explanation has been written in the manuscript, marked in red at L99-105.

  1. Line 98: please discuss about the relation between pseudoplastic behavior of KGM and the rheology behavior requirement of dysphagia liquid food.

Response: We have added the corresponding description and marked the revised portion in red in the manuscript (L105-109).

  1. Lines 112-115: Please explain more how to know that the increase of temperature did not eliminate the hydrogen bonding between KGM and water.

Response: I'm sorry that the description of this content was not accurate enough. We have revised it in the manuscript and marked in red. (L130-133)

  1. Figure 4: Please discuss about tan d (G”/G’) of samples to confirm the gel like behavior. Tan d in the range of 0.1-1 is suitable for dysphagia liquid food.

Response: The tan δ of the samples has been discussed in Lines 319-326 of the manuscript and illustrated in Figure 13c. The corresponding parts were marked in red.

  1. Line 137: Could you please give the information of melting point temperature of KGM?

Response: We are very sorry that we found the melting point parameter has little to do with the thickening properties involved in the study, so we have deleted this part in the manuscript.

  1. Line 176: please give the reference showed that high temperature degrade GG.

Response: The reference is as follow: T.D. Bradley, A. Ball, S.E. Harding, J.R. Mitchell. Thermal degradation of guar gum. Carbohydrate Polymers. 1989, 10(3), 205-214. [https://doi.org/10.1016/0144-8617(89)90012-X]. And it has been added to the manuscript. (L190)

  1. Why was pure KGM not suitable for using as thickener in dysphagia food? Why was the thickener formula needed to develop?

Response: Although pure KGM solution can meet the viscosity requirements of foods with swallowing disorders, it can only provide a single viscosity, and cannot maintain stability under complex conditions such as temperature, shear behavior and set time. Meanwhile, the taste of pure KGM is a little rough and the dissolvability is not very good, so we choose a variety of thickeners and maltodextrin compound to prepare dysphagia food.

  1. Line 228: According to Figure 8c, the case of LBC showed low viscosity compared to other cases. Please give the reason for the sentence “LBG could make the viscosity of the compound thickener…”.

Response: Among the compound thickeners, XG has the highest viscosity, and the gel is formed after heating and cooling, which reduces the fluidity and is not conducive to swallowing. The addition of LBG will appropriately reduce the viscosity, and LBG has the property of complete hydrolysis at 80 ℃, and the viscosity of LBG solution will remain stable after a small increase after cooling.

  1. Lines 305-312: Which shear rate was used to monitor the viscosity and consistency type of samples? Which shear rate is conventionally used to monitor the viscosity of foods during swallowing?

Response: We were based on dysphagia guide published by National Dysphagia Diet Task Force & American Dietetic Association. Dysphagia guide classifies foods according to their shear viscosity at a shear rate of 50 s-1 at 25 ℃, at the following levels: thin (1–50 mPas); nectar-thick (50–350 mPas); honey-thick (350–1750 mPas); pudding-thick (>1750 mPas).

  1. Please show the FTIR results for studied thickener and thickening component to confirm synergistic of components.

Response: We have completed the FT-IR testing, and the relevant analysis and method have been written in parts 2.4.4 and 3.5.4 of the manuscript. (L393-417, L541-545)

  1. Line 415: Please give the detail of shaking speed and time for making fully swollen.

Response: We have revised it in the manuscript at L454, 460, 472, and 482.

  1. Lines 423-425: please give the information of design of experiment (DOE) method.

Response: KGM is the basic material and has good synergistic effect with GG and XG, so the mass ratio of KGM is the highest. XG has a lower mass ratio because it is too viscous and forms irreversible gels after cooling. when LBG and KC were mixed with KGM, the viscosity of the compound solution would be relatively reduced. In addition, experimental groups without LBG and KC were set up to determine their role in the compound thickener. We have added an explanation in L467-469 of the manuscript.

  1. Line 432: Why was the compound thickener of KGM:GG:XG:LBG:KC = 13:2:2:2:1 chosen for studying the effect of maltodextrin?

Response: In 2.2.1, we explored the trend of viscosity change of the thickeners mixed with different ratios and analyzed the properties of the five thickeners. At the ratio of 13:2:2:2:1, the compound thickener has moderate viscosity, can be completely dissolved in water with good fluidity, and will not form gel after heating. Therefore, 13:2:2:2:1 was selected as the optimal ratio to study the effect of maltodextrin.

  1. Issues 3.3 and 3.4: What is the mixing method of component of thickener? Is it the physical mixing?

Response: Yes, component of thickener was made by physical mixing. (L458-459)

  1. Table 2: Could you please give the information of the range of experimental condition design?

Response: In the single factor experiments taking sensory evaluation as an indicator (issue 3.4), we considered 10.0 mg/mL as the optimal concentration of maltodextrin, and the compound thickener mass concentrations of 5.0, 6.0, and 7.0 mg/mL were inferred to meet the requirements of different patients. On this basis, the thickener mass concentration in 1# is 5 mg/mL, and the concentration of maltodextrin is 10 mg/mL. By analogy, the content of each component of thickening component in Table 2 can be obtained.

    Thank you very much for your comments and suggestions.

Looking forward to hearing from you.

Thank you and best regards.

Yours sincerely,

Wen Zhang

March. 19, 2022

Reviewer 4 Report

This manuscript provides a technical reference for KGM as dietary nutrition support combined with guar gum (GG), xanthan gum (XG), locust bean gum (LBG), carrageenan (KC) for dysphagia patients. The rheological measurements and sensory evaluation were applied as an indicator to optimize the formulation and preparation processes of thickening component. The description is generally easy to read. However, some further clarifications should be done before being accepted.

Comments:

The rheology section should be more elaborated.

L37: “reduce the risk of swallowing”: did you mean “reduce the risk of aspiration during swallowing”or “strangulation” or some other things?

L44: Fig. 1 is not clear enough to publish.

L58: “Carrageenan (KC)”: “κ- Carrageenan” ?or some other types?

L105: Figures 2, 3, 4, 5, 7, 8, 9, 12, 13, and 14: it is better to adopt log-log plots describe the variation of these rheological parameters.

L314: some bisible vesicles and heterogeneous component can be seen in your samples as shown in Fig. 15. When the authors conducted rheological measurements, did you exclude the influence of vesicles on the experimental results? In other words, were these experimental results repeatable?

L366: in Fig. 18, impure peaks should be eliminated or explained clearly for the XRD patterns.

L532: Please check the references carefully and unify their format. Including the doi, the abbreviation or full name of journal and writer’s name. For examples, there is no “doi” in ref 1; ref 5 and 9 share the same first writer, but the name show difference in expression: “Cichero J” and “Cichero, J.A.”, and so on. Apart from that, the number of reference is not enough to be a good work.

Author Response

Dear Editors and Reviewers,

     Thank you for your letter and comments concerning our manuscript entitled “Preparation and Performance of Thickened Liquids for Patients with Konjac Glucomannan Mediated Dysphagia” (ID: molecules-1598956). Those comments are all valuable and helpful for revising and improving our paper. We have studied comments carefully and have made correction which we hope meet approval. Revised portion were marked in red in the paper. The main correction in the paper and the response to the reviewers’ comments are as following:

  1. The rheology section should be more elaborated.

Response: We have described in detail the relationship between shear rate and shear stress in the rheology section of the samples, and analyzed the tan δ of the samples, which were marked in red at L295-299 and L316-326 in the manuscript respectively.

  1. L37: “reduce the risk of swallowing”: did you mean “reduce the risk of aspiration during swallowing” or “strangulation” or some other things?

Response: Yes, it’s “reduce the risk of aspiration during swallowing”. We have redescribed it in the manuscript and marked it in red. (L35-38)

  1. L44: Fig. 1 is not clear enough to publish.

Response: It may be unclear due to the previous image format, and I re-uploaded the figure with a higher resolution.

  1. L58: “Carrageenan (KC)”: “κ- Carrageenan” ?or some other types?

Response: Carrageenan is supplied by the manufacturer and is a compound of κ- Carrageenan and potassium chloride.

  1. L105: Figures 2, 3, 4, 5, 7, 8, 9, 12, 13, and 14: it is better to adopt log-log plots describe the variation of these rheological parameters.

Response: Thank you for your advice. We have been modified Figure 2, 3, 4, 11, 12, and 13 to be log-log plots. In addition, Figure 6, 7, and 8 (the figure numbers have been changed, these are the latest numbers) mainly describe viscosity change process, including peak viscosity, valley viscosity and final viscosity. And I sincerely think that linear plots would better reflect the specific viscosity values.

  1. L314: some bisible vesicles and heterogeneous component can be seen in your samples as shown in Fig. 15. When the authors conducted rheological measurements, did you exclude the influence of vesicles on the experimental results? In other words, were these experimental results repeatable?

Response: Before the rheological measurements, the samples were ultrasonic for 10 min, in which the visible vesicles were significantly reduced and the tiny bubbles were more uniform, so as to minimize the influence of bubbles on the rheological measurement results. Thus, the experimental results have good repeatability. Moreover, we have changed Figure 14 (the figure number has been changed, this is the latest number) in the manuscript.

  1. L366: in Fig. 18, impure peaks should be eliminated or explained clearly for the XRD patterns.

Response: The impure peaks in the XRD patterns have been smoothed out. In Figure 17 (the figure number has been changed, this is the latest number), the main peak positions of the raw material powders and the thickening components are approximately the same, aiming to show that the thickening components are made in accordance with the process requirements.

  1. L532: Please check the references carefully and unify their format. Including the doi, the abbreviation or full name of journal and writer’s name. For examples, there is no “doi” in ref 1; ref 5 and 9 share the same first writer, but the name show difference in expression: “Cichero J” and “Cichero, J.A.”, and so on. Apart from that, the number of reference is not enough to be a good work.

Response: Thank you very much for your valuable advice. The references have been changed to a uniform format, and the number has increased to 49.

    Thank you very much for your comments and suggestions.

Looking forward to hearing from you.

Thank you and best regards.

Yours sincerely,

Wen Zhang

March. 19, 2022
